# Celcomen: spatial causal disentanglement for single-cell and tissue perturbation modeling

Stathis Megas [1,2,3,4,5,6,10] ✉, Daniel G. Chen [1,2,7,10], Krzysztof Polanski[1,2], Hesam Asadollahzadeh[1], Moshe Eliasof[8], Carola-Bibiane Schönlieb[8] & Sarah A. Teichmann [1,2,3,9] ✉

Celcomen leverages a mathematical causality framework to disentangle intra- and inter-cellular gene regulation programs in spatial transcriptomics data through a generative graph neural network. It is a first step towards perturbation models of Virtual Tissues and can generate post-perturbation counterfactual spatial transcriptomics, thereby offering access to experimentally inaccessible samples. We validated its disentanglement, identifiability of causal structure, and counterfactual prediction capabilities through simulations and in clinically relevant human glioblastoma, human fetal spleen, and mouse lung cancer samples. Celcomen provides the means to model disease- and therapy-induced changes allowing for new insights into single-cell spatially resolved tissue responses.

A cell's gene expression profile simultaneously encodes information about its intrinsic characteristics and extrinsic tissue microenvironment. Recent technologies now allow for large-scale profiling of transcriptomes at single-cell resolution with spatial context[1–4]. With these technological advances, computational methods that can disentangle intrinsic versus extrinsic inter-cellular regulation of gene expression are needed. These disentangled representations are necessary to fully reconstruct the complex interplay of intra- and inter-cellular interactions in human tissues during homeostasis and post-disease or therapy induced perturbation[5,6].

Several previous works relied on prior knowledge of protein-protein interactions or gene regulatory networks to distinguish intrinsic and extrinsic circuits; this reliance often excludes key cell-cell interaction partners that are unreported[7,8]. Recent deep learning models advance on this limitation by simultaneously modeling intrinsic and extrinsic features; however, these models lack interpretable insight due to their black box nature[9]. Further, most current models lack mathematical guarantees, leading to their hypersensitivity to input data variability; exceedingly few accept both spatial and single-cell input data[10,11]; and many cannot perform in silico perturbation experiments critical to understanding tissue behavior during disease[12–14]. While these works have introduced marked computational leaps in spatial transcriptomics data interpretation, they often cannot perform causal inferences due to their lack of identifiability which mathematically prevents many current models from deriving comprehensive mechanistic insights into cell and tissue biology.

Celcomen overcomes some of these limitations by leveraging an identifiable causal structure learning framework into a generative graph neural network for learning disentangled representations of intra- and inter- cellular gene regulation in spatial transcriptomics data (Fig. 1a, b). The inference module of Celcomen, hereby called CCE, finds disentangled representations of gene interactions at the cell or 1st neighbor levels with focus placed on paracrine and autocrine

[1]Department of Cellular Genetics, Wellcome Sanger Institute, Hinxton, UK. [2]Department of Medicine, Cambridge Stem Cell Institute, University of Cambridge, Cambridge, UK. [3]Department of Applied Mathematics and Theoretical Physics, Cambridge Center for AI in Medicine, University of Cambridge, Cambridge, UK. [4]Department of Medicine III, Medical University of Vienna, Vienna, Austria. [5]MetAGE Cluster of Excellence, Vienna, Austria. [6]Center for AI in Medicine, Medical University of Vienna, Vienna, Austria. [7]UCLA-Caltech Medical Scientist Training Program, University of California, Los Angeles, CA, USA. [8]Department of Applied Mathematics and Theoretical Physics, University of Cambridge, Cambridge, UK. [9]Canadian Institute for Advanced Research, Toronto, ON, Canada. [10]These authors contributed equally: Stathis Megas, Daniel G. Chen. ✉e-mail: stathis.megas@meduniwien.ac.at; sat1003@cam.ac.uk

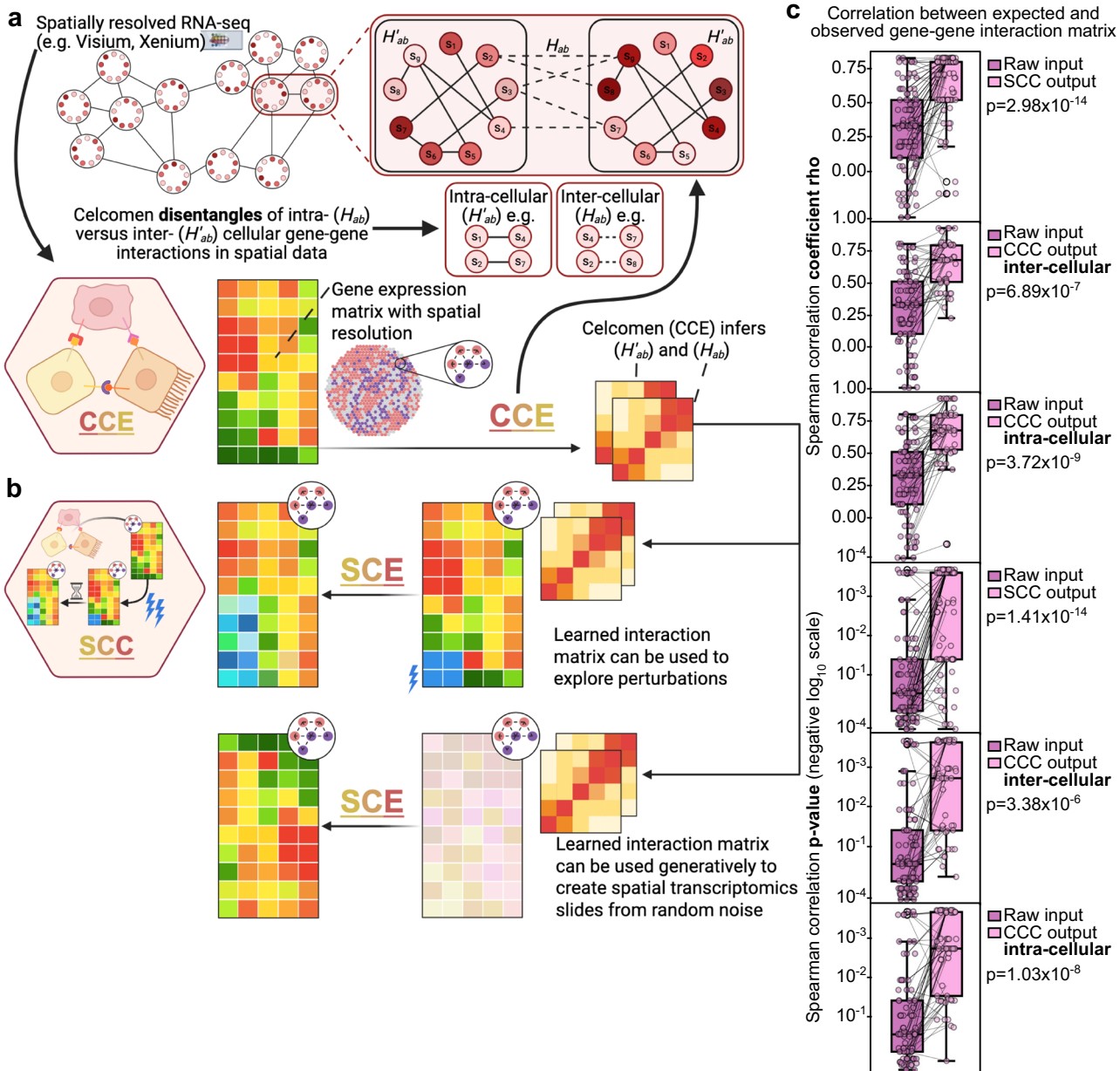

**Fig. 1 | Celcomen reproduces its identifiability guarantees in simulations.**
**a** Celcomen (CCE) can learn gene-gene relationships from spatially resolved data. The highlighted cell-cell pair in spatial data, emphasizes how CCE can distinguish gene-gene interactions that are intra- ($H'_{ab}$) vs. inter- ($H_{ab}$) cellular. **b** Simcomen (SCE) leverages learned gene-gene relationships from CCE to model tissue behavior after cellular or genetic perturbation. SCE also possesses generative properties through its ability to create tissue-condition representative spatial transcriptomics data given an established matrix of gene-gene relationships. **c** Box plots with x-axis as the comparison, in detail, in magenta we compare the random count matrix with

the ground truth and in light pink we compare the learned count matrix (SCE output) or gene-gene interaction matrix (CCE output) with ground truth. Y-axis is Spearman correlation coefficient rho (upper three) or $p$-value (lower three). Mann–Whitney U-test $p$-values are labeled on the center right of each plot and the legend on the upper right of each plot labels each box's dataset. Each dot represents an individual simulation using a given random seed ($n = 100$ simulations were performed for each dataset). Box plots represent median and interquartile range. Source data are provided as a Source Data file. Created in BioRender. Chen, D. (2026) https://BioRender.com/szumz0e.

signaling. These representations can then be used by the generative module of Celcomen (Simcomen), hereby called SCE, to produce single-cell spatially resolved predictions of tissue behavior post perturbation and to derive realistic slides of spatial transcriptomics data from noise. We validated the robustness and insightfulness of CCE and SCE across a plethora of simulations, and input types, including in human tissues. In summary, we demonstrate Celcomen as a mathematically grounded spatial and single cell transcriptomics analysis tool that introduces the capability to perform high-resolution spatially resolved perturbation predictions that are critical for clinically relevant disease modeling and tissue engineering efforts.

## Celcomen's mathematical identifiability guarantees are reproduced in simulations

Causal inference frameworks seek to uncover the mechanisms that generate the observed data, and to ensure this, they rely on the mathematical principle of identifiability[15–18]. This principle holds true when there exists a single unique model and parameter combination that fits the data, and thus we are assured that our observations can be explained by the given model. However, despite the useful properties that are enjoyed by these identifiable models (e.g., robustness, generalizability, and self-consistency), most current deep learning models violate this principle[16,19]. We overcome this limitation through

mathematical proofs of Celcomen's identifiability of causal structure (see Supplemental Notes).

To confirm that Celcomen's identifiability guarantees exist in practice, we subjected Celcomen to a multitude of self-consistency simulations (see "Methods"). First, we randomly generated a ground truth set of gene-gene interactions. Next, we utilized Celcomen's generative module, SCE, to generate spatial transcriptomics data representative of these gene-gene interactions. Then, we fed the generated data into Celcomen's inference module, CCE, in an attempt to retrieve the originally encoded gene-gene interaction forces. Celcomen consistently demonstrated strong alignment between its inferred gene-gene interactions from its simulated data and the ground truth (Fig. 1c). This suggests that Celcomen is self-consistent and identifiable, as it appears to be able to move between encoded gene-gene interactions to simulated spatial transcriptomics and then back to inferred gene-gene interactions with seemingly minimal loss of information.

To validate Celcomen's predictions on ex vivo human data, we applied our model to multiple spatial transcriptomics slides of human fetal spleen[20]. For each slide, we trained a sample-specific model and a model trained on the remaining samples. We then correlated the gene-gene interaction matrices of these two models. In line with its claimed identifiability, we observed strong positive correlation between these two gene-gene interaction matrices even though they shared no training samples (Supplementary Fig. 1). Interestingly, Celcomen finds strong gene-gene interactions between IFNGR1/2, JAK1/2, STAT1 and MX1. These genes compose a well known signaling pathway whereby IFNG binds to the IFNGR1/2 dimer whose signal is transduced by the JAK1 and JAK2 to STAT1 with MX1 as the downstream target[21]. So although Celcomen does not have the degrees of freedom to capture multi-gene interactions, it does capture that MX1 is regulated by this canonical pathway.

## Celcomen recapitulates expected immune programs upon interferon perturbation

Having validated Celcomen's robustness as a model through simulation-based testing (Fig. 1), we then sought to test its value and validity in disentangling intra- versus inter- cellular gene regulation programs and in performing spatially resolved perturbation modeling. To test these claims, we applied Celcomen in human clinical setting by analyzing a single-cell resolution spatial transcriptomics dataset of human glioblastoma (brain cancer) (Fig. 2a). Consistent with its core theory, we found that Celcomen was able to successfully disentangle intrinsic versus extrinsic sources of transcriptomic variation through its assignment of gene-gene interactions involving genes encoding secreted proteins or factors as inter-cellular, and those solely involving cytoplasmic genes encoding cytoplasmic signaling proteins as intra-cellular (Fig. 2b). It is important to note that the knowledge of which genes are encoding secreted proteins and which cytoplasmic ones is not provided into the model as prior information, but rather is learned by the model in an unsupervised manner.

We leveraged Celcomen's perturbation abilities to model interferon signaling in the context of a neurological tumor, where we investigate the scenario of interferon knockout. We chose to model interferon signaling due to its critical role in cancer in inducing antigen presentation, inflammation, and immune activation[22–24]. First, we quantified the expression of our sample's interferon associated gene program by averaging differentially upregulated genes in interferon (IFITM3) high versus low cells. Next, we knocked out interferon expression in a randomly chosen interferon high cell (Fig. 2c). Utilizing this interferon score, we not only confirmed our in silico knockout of interferon in the perturbed cell, by observing its marked loss of interferon associated genes, but we also observed loss of interferon signaling in neighbors of the perturbed cell (Fig. 2d). This behavior is highly consistent with known interferon biology as interferon signaling

physically propagates from cell-to-cell within human tissues; thus recapitulating this intercellular signaling phenomenon supports the validity of Celcomen's perturbation modeling[25–27].

To further confirm the validity of our interferon knockout modeling, we performed pathway enrichment on genes that were differentially changed in perturbed (and perturbed neighboring) compared to unperturbed cells (see "Methods"). Indeed, we find that post interferon knockout, perturbed cells and their neighbors significantly downregulated characteristic interferon response programs compared to unperturbed cells (Fig. 2e). For example, we observed the perturbed cells to have decreased T cell effector and activation gene programs, as well as greater loss of infection-related gene sets and marked increases in regulatory programs; we note that some gene sets were driven by non- cell type specific genes e.g., CCL2 and CCR1 for immune effector programs. The consistency of our model with multiple aspects of known interferon biology strongly affirms Celcomen's ability to model perturbations with spatial resolution.

Moreover, to demonstrate Celcomen's ability to learn differential gene regulation and cell communication between health and disease, we curated a paired ST dataset of the human brain from a healthy donor and patient with glioblastoma. We then trained separate Celcomen models for each ST dataset and extracted each model's intra- and inter- cellular gene-gene interaction matrices to identify interaction programs differential between the healthy brain and one with glioblastoma (GBM) (Supplementary Figs. 2, 3). Celcomen disentangled gene programs that were shared between healthy and diseased brain indeed involved core neuronal functions, such as intra-cellular cluster 4 and inter-cellular cluster 2 being enriched for nervous system and synaptic processes (Supplementary Figs. 2d, 3d). For this glioblastoma example, we also identify an increase in inflammation, likely representing the immune response to the tumor but we also potentially identify inappropriate downregulation of cell clearance that may impede proper neuronal function and aid tumorigenesis[28].

Thus, through an in-depth case study of Celcomen on ex vivo human tissue sections, we provide significant validation for disentangling intra- versus inter-cellular gene regulation programs, in learning differential programs between health and disease, and in performing accurate high-resolution spatially contextualized perturbation modeling.

## Celcomen spatial perturbation predictions are validated in vivo

Having validated Celcomen with ex vivo human tissue, we sought to in vivo validate its spatial perturbation modeling ability using published spatial transcriptomics on genetically perturbed tumor lesions from a mouse model of human lung cancer (Fig. 3a)[29]. We note the only available platform, as of now, for spatial CRISPR perturbations is via the 10x Genomics Visium platform which has a resolution of 1–30 cells per spot[30,31]. The ideology behind this in vivo validation was to (1) train on lesions not specific to one gene (KP), (2) simulate a one gene specific genetic perturbations in KP tissue, (3) compare model predicted transcriptomic differences with experimentally observed differences. To achieve this, we first isolated KP annotated lesions not specific to one gene from the Visium slide and removed any spots proximal to experimentally perturbed spots, this mitigates information leakage issues (Fig. 3b, see "Methods"). We used Celcomen to identify lung cancer specific gene-gene interaction modules from the WT lesions and then leveraged our model's generative component to predict spot-resolution transcriptomic profiles upon in silico Tgfbr2 knockout (KO). Consistent with Celcomen's accuracy in ex vivo human tissue, we once again observed strong agreement, significant positive correlation, between model predicted and experimentally observed transcriptomic changes, comparing Tgfbr2 KO and WT spots (Fig. 3c). Through random permutation experiments, we confirmed that these correlations were unlikely to occur by chance which strongly supports

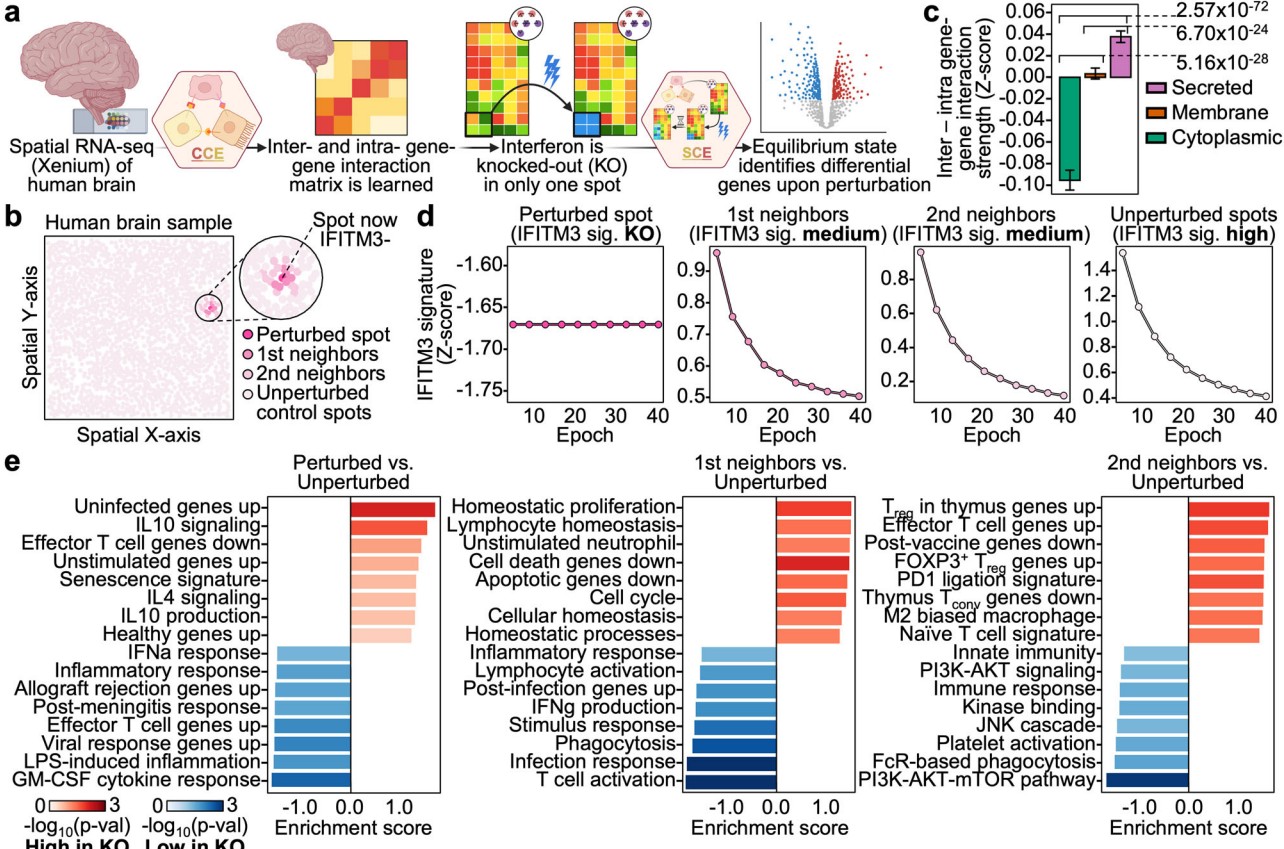

**Fig. 2 | Celcomen recapitulates known interferon knockout biology in human glioblastoma and disentangles intra- and inter- cellular gene-gene interactions. a** Public spatially resolved RNA-seq data, Xenium, of the human brain during glioblastoma is inputted into CCE to derive intra- and inter-cellular gene-gene relationships. Interferon (IFN) signaling is in silico knocked-out (KO) in a previously IFN⁺ cell and SCE learns the local and global effects of this perturbation. **b** Cells within the region of interest are plotted with their spatial x- and y- coordinates. Legend on the lower right labels the identity of the perturbed cell in dark pink and its most proximal (1st) and lesser proximal (2nd) neighbors in lighter pink shades. **c** Bar plot with the x-axis as the subcellular localization of the gene as acquired from its gene ontology and the y-axis is the difference between the gene's inter- and intra-cellular gene-gene interaction terms. Mann–Whitney U-test p-values are labeled on the plot, bar height represents mean and, error bars denote 95% confidence intervals. Each dot represents an individual gene-gene interaction; there are

$n = 132,000$ interactions involving membrane associated genes, $n = 120,960$ for secreted, and $n = 51,840$ for cytoplasmic. **d** Scatter plots with the x-axis as the epoch number and the y-axis as the interferon signature score of the given spot(s) at the specified epoch. Identity of the spot(s) of interest are labeled at the top of each plot and spot colors match those in (**b**). **e** Bar plots with the color as the pathway enrichment significance, see legend on the lower left, the x-axis as the enrichment score, and the y-axis for pathway names. Pathways were derived by first calculating pre- and post- perturbation changes in gene expression in each cell, then identifying differentially changed genes between spot(s) of interest and unperturbed controls, this provides a ranking of genes that were differentially upregulated or downregulated in the interferon KO cell, or its neighbors, as compared to the unperturbed control cells. Source data are provided as a Source Data file. Created in BioRender. Chen, D. (2026) https://BioRender.com/0051dkf.

Celcomen's ability to model spatial perturbations in a manner that agrees with experimentally derived ground truth (Fig. 3d). In further support, we repeated this validation by comparing in vivo Jak2 KO and Celcomen in silico Jak2 KO spots with WT and, once again, observed significant positive correlation between Celcomen predicted and experimentally observed transcriptomic changes (Fig. 3e, f). Thus, not only are we able to validate Celcomen's ability to recapitulate known biology in human tissue, but we are also able to in vivo validate Celcomen's spatial perturbation modeling capabilities in clinically relevant models of human disease.

## Discussion

The advent of single-cell resolution spatial transcriptomics has brought about a new paradigm to human cell mapping, allowing for spatial tissue atlases with unprecedented resolution[29–33]. While many computational methods have begun to address the phenotypic characterization of spatial transcriptomics[10], Celcomen takes the next step and performs tissue level perturbation modeling. There is a critical need for these methods in order to understand the mechanisms

behind tissue dysfunction during disease states. Current works that address this need are often uninterpretable, with putatively causal mechanisms hidden with a black box, or they are not mathematically robust leading to high variance in model outputs that limit their use.

Celcomen is a perturbation modeling framework of tissue states with spatial context, while also providing highly interpretable results, through its disentanglement of cell intrinsic versus extrinsic gene regulation programs, and mathematical robustness through an identifiability guarantee. We confirm Celcomen's ability to disentangle and recover ground truth gene-gene interactions in real and self-simulated spatial transcriptomics data. These multi-faceted advances of Celcomen are likely to provide actionable insights into how human diseases cause tissue failure and allow for new testable hypotheses into the ways in which therapies provide patients with real clinical benefit. We anticipate that as technology continues to advance, the value of Celcomen and its future iterations will only continue to grow as it becomes more feasible to model disease state and more important to understand how.

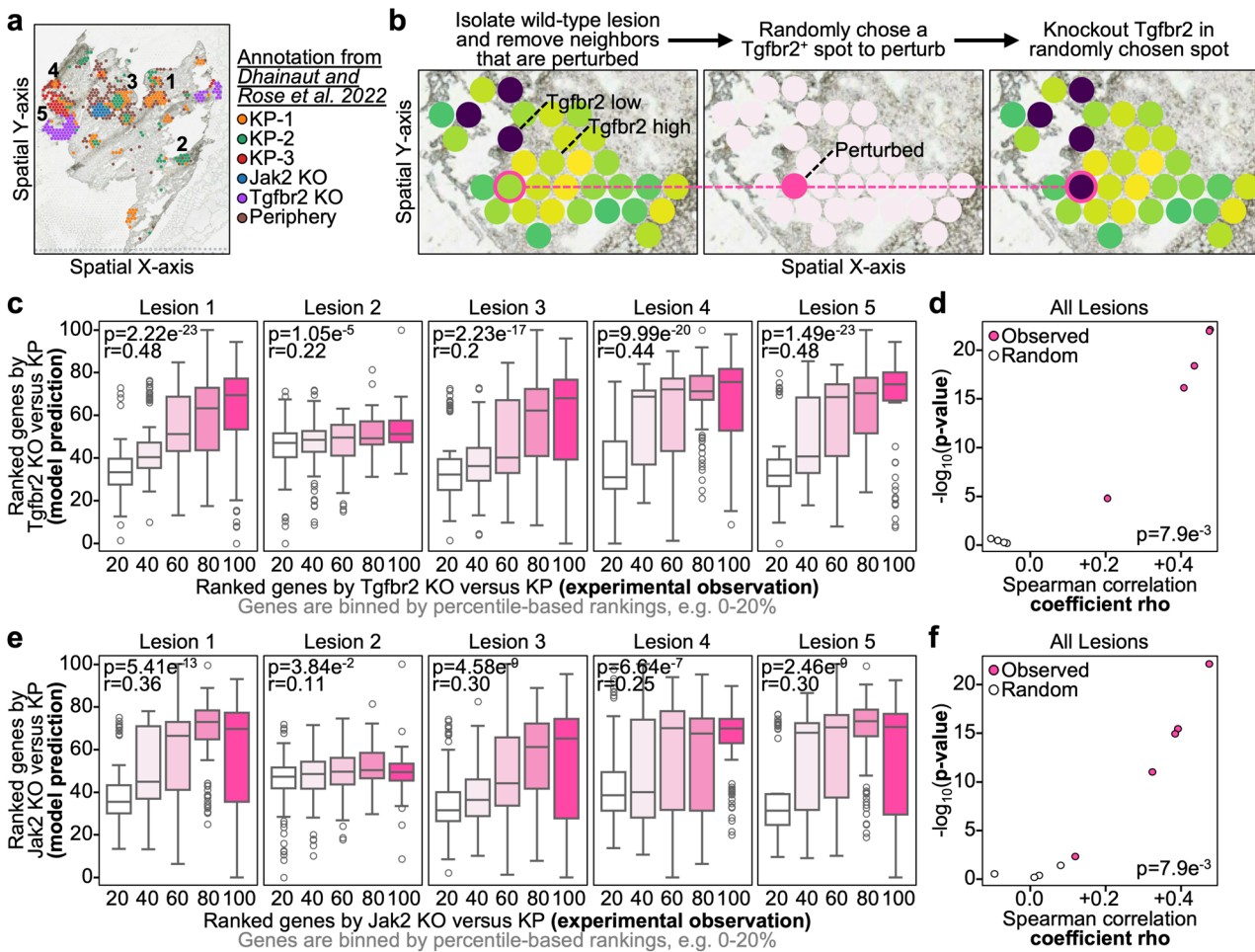

**Fig. 3 | Celcomen counterfactual predictions are validated in vivo in a clinically relevant lung cancer model. a** Scatter plot on spatial axes with dots representing tumor lesions, color represents tumor cell phenotype, perturbation specific clusters are labeled with "KO" for knockout and tumors not specific to one gene KO are labeled with "KP", see legend on right. Lesions of interest, large enough for modeling, are labeled with numbers. **b** Example workflow with a KP lesion not specific to one gene KO (KP), and thus used as background, trimmed for spots within two-degrees of perturbed clusters, random Tgfbr2$^+$ spot has Tgfbr2 knocked out, our model then predicts the whole transcriptome that accompanies this perturbation. **c** Box plots, per lesion, with x-axis as the observed ranked differentially expressed genes (DEGs) between Tgfbr2 KO and KP and the y-axis as the model predicted gene ranking between our perturbed Tgfbr2 KO spot and KP spots. Spearman correlation coefficient rhos and *p*-values are annotated on the plot. Each dot represents an individual gene (*n* = 377 differentially expressed genes). Box plots represent median and interquartile range. **d** Scatter plot with each dot representing

a given tumor lesion with Tgfbr2 KO and the x-axis as the Spearman correlation coefficient rho and y-axis as the *p*-value, the color indicates if the correlation was computed on the lesion's observed gene rankings or a random shuffling of the gene rankings. Mann–Whitney U test *p*-value between observed and randomly shuffled correlations are annotated on the plot. **e** Box plots, per lesion, with x-axis as the observed ranked differentially expressed genes (DEGs) between Jak2 KO and WT and the y-axis as the model predicted gene ranking between our perturbed Jak2 KO spot and wild type spots. Spearman correlation coefficient rhos and *p*-values are annotated on the plot. **f** Scatter plot with each dot representing a given tumor lesion with Jak2 KO and the x-axis as the Spearman correlation coefficient rho and y-axis as the *p*-value, the color indicates if the correlation was computed on the lesion's observed gene rankings or a random shuffling of the gene rankings. Mann–Whitney U test *p*-value between observed and randomly shuffled correlations are annotated on the plot. All *p*-values were calculated in a two-tailed fashion. Source data are provided as a Source Data file.

Moreover, Celcomen aims to identifiably retrieve the undirected form of the DAG underlying intra- and inter-cellular regulation from observational data. Without interventional data, it is not possible (without further assumptions) to uncover more than the Markov equivalence class of the DAG, which is the undirected form of the graph along with partial information about colliders, which hint towards directionality. Therefore, Celcomen's ability could be improved by training in a supervised fashion on spatial transcriptomics data with perturbations, which might soon be available in higher volumes.

Finally, although technically not a limitation, we want to explain that for Celcomen to be able to disentangle intra- and inter-cellular gene regulation, it needs data that are of single-cell resolution. When run on multi-cell resolution data, such as Visium (at 55 μm resolution),

Celcomen disentangles intra-spot and inter-spot gene regulation. Intra-spot gene regulation would involve signatures of intra-cellular, juxta-crine, and short-range paracrine gene regulation, whereas inter-spot gene regulation would involve long-range paracrine gene regulation.

## Methods

### Model: Theoretical foundations

We provide here a brief overview of the theory and architecture of Celcomen, for more details see Supplementary Note 1. Celcomen estimates for each spatial transcriptomics sample $\{s^{\alpha}_i : \alpha = 1, ..., G; i = 1, ..., N\}$, where N is the number of cells and G is the number of cells in the sample, the joint conditional probability $p(\{s^{\alpha}_i\}|g_{\alpha,\beta}, g'_{\alpha,\beta})$, where g and g' are the undirected graphs underlying

the directed causal model graphs (see Fig. 1). We assume here that there is a global causal model that governs gene regulation within and cell communication between all cell types in a specific context such as tissue, disease state etc. This distribution is estimated using a Generative Graph Neural Network during training, and it can then be prompted with perturbations to predict spatial counterfactual outcomes such as what the spatial gene expression would have been had e.g., a gene been knocked-out in one or more cells. The synthetic data generation in Celcomen is mechanistically grounded in the aspects of the causal model it identifies during training. Grounding the data generation on a causal model not only confers robustness, but also aids interpretability by disentangling the effects of genes into intracellular and intercellular.

To our knowledge Celcomen is the first *perturbation model* of Virtual Tissues that enables prediction of spatial counterfactuals and has been benchmarked on in-vivo datasets, fulfilling an important gap in the literature.

The entire neural network architecture of Celcomen is derived from its three main assumptions that:

- Celcomen's expected gene-gene correlations within the same cell in its synthetic data should match exactly the empirical ones
- Celcomen's expected gene-gene correlations within two neighboring cells in its synthetic data should match exactly the empirical ones
- Causal sufficiency that there are no unobserved confounders that affect two or more genes.

Mathematically phrased we demand that the model should minimize the entropy $S(p(\{s_i^\alpha\}), g_{\alpha\beta}, g_{\alpha\beta}) = -\sum_{\{s_i^\alpha\}} p(\{s_i^\alpha\}) \log(p(\{s_i^\alpha\}))$

$$+ \sum_{\alpha,\beta} g_{\alpha\beta} \left( <\sum_{i,jnn} s_i^\alpha s_j^\beta>_P - <\sum_{i,jnn} s_i^\alpha s_j^\beta>_{emp} \right)$$

$$+ \sum_{\alpha,\beta} g'_{\alpha\beta} \left( <\sum_i s_i^\alpha s_i^\beta>_P - <\sum_i s_i^\alpha s_i^\beta>_{emp} \right), \quad (1)$$

where $g_{\alpha,\beta}, g'_{\alpha,\beta}$ are the Lagrange multipliers enforcing our assumptions as constraints into the model. It is a well known fact from theoretical physics that the Lagrange multiplier of a constraint is equal to the force that implements the constraint, and as such is deconfounded – for instance, in the context of Gaussian graphical models the Lagrange multipliers are equal to the partial correlations.

Celcomen is a robust algorithm for maximizing this entropy functional to learn the joint probability distribution $p(\{s_i^\alpha\})$. Celcomen's entire architecture is mathematically derived from Eq. 1, which severely limits the arbitrary choices we had to make in designing Celcomen to a few hyperparameters for which we provide guidance. The derivation of Celcomen's architecture is the content of the following proposition.

**Proposition 1**. The following two optimization problems are equivalent:

- Maximizing $S(p(\{s_i^\alpha\}), g_{\alpha\beta}, g_{\alpha\beta})$ over all possible functions $p \in L^1(R^{N \times G})$ and matrices $g_{\alpha,\beta}, g'_{\alpha,\beta}$
- Minimizing
$min_{g,g'} <\log p>_{emp} = min_{g,g'} \left( - \log Z(g_{\alpha\beta}, g_{\alpha\beta}) + g_{\alpha\beta} C^{emp}{}_{\alpha\beta} + g_{\alpha\beta} C'^{emp}{}_{\alpha\beta} \right)$

where $C^{emp}{}_{\alpha\beta} = \sum_{i,j} s_i^\alpha J_{ij} g_{\alpha\beta} s_j^\beta$, $C'^{emp}{}_{\alpha\beta} = \sum_i s_i^\alpha g_\alpha, \beta' s_i^\beta$, $Z = \sum_{\{s_i^\alpha\}} e^{H(\{s_i^\alpha\})}$ and $H = \sum_{\alpha,\beta} s_i^\alpha g'_{\alpha\beta} s_i^\beta + \sum_{\alpha,\beta} \sum_{i,j} s_i^\alpha J_{ij} g_{\alpha\beta} s_j^\beta$.

Sketch of proof: we demand that the functional derivative $\frac{\delta S(p(\{s_i^\alpha\}), g_{\alpha\beta}, g'_{\alpha\beta})}{\delta p(\{s_i^\alpha\})} = 0$, we then solve this equation to find $p(\{s_i^\alpha\})$ in terms of $g_{\alpha\beta}, g'_{\alpha\beta}$, and then substitute this in $S(p(\{s_i^\alpha\}), g_{\alpha\beta}, g'_{\alpha\beta})$. For full details we refer the reader to the supplementary material.

To efficiently compute the partition function and accelerate the training process we derive and implement an approximation to the partition function Z (see supplementary material).

The fundamental limitation of observational data is that it can be used to learn only certain aspects of a causal model. For instance, in linear models it can be shown that to fully identify a generic causal model with n nodes we need n-1 interventions[34]. Indeed, observational data can identify the causal model only up to its Markov equivalence class, which is the undirected underlying graph with some information about colliders that hint towards directionality[35]. In accordance with the causal inference theory, we confirm Celcomen's inability to infer information more fine grained than the Markov equivalence class of the graph in simulations (Supplementary Fig. 4).

In the following proposition we show that Celcomen can identify the undirected underlying graph, in the sense that each time there is a unique underlying undirected graph that fits the data optimally. Therefore, Celcomen's ability is almost tight against the best possible bound on the abilities of unsupervised models.

**Proposition 2**. The model defined by $p(\{s_i^\alpha\}|g_{\alpha\beta}, g'_{\alpha\beta}) = \frac{e^{H(s_i^\alpha)}}{Z}$ is identifiable in the sense that

$\forall \{s_i^\alpha\} : p(\{s_i^\alpha\}|g_{\alpha\beta}, g'_{\alpha\beta}) = p(\{s_i^\alpha\}|h_{\alpha\beta}, h'_{\alpha\beta}) \Rightarrow g_{\alpha\beta} = h_{\alpha\beta}$ and $g'_{\alpha\beta} = h'_{\alpha\beta}$.

Sketch of proof: we take derivatives with respect to a gene's expression, and then set it equal to zero. For full details we refer the reader to the supplementary material.

One important property of Celcomen, supported by Proposition 2, is its ability to reduce confusion from confounders and mediators when estimating causal effects. Even when two genes show a high correlation without a direct causal link, Celcomen's gene-gene force term (the Lagrange multiplier) tends toward zero, helping the model to distinguish genuine causal interactions from indirect associations. To illustrate the scale of the challenge, consider that a typical cell may express on the order of $10^4$ genes and interact with 10 neighbors; as a result, any direct effect between two genes could be masked by $10^5$ indirect effects mediated through one gene, or $\sim 10^{10}$ indirect effects mediated through two genes, etc. Celcomen aims to identify the subset of direct effects that best explain the observed data while accounting for such infinite layers of mediation and confounding.

There is a beautiful story of how theorems in mathematics and discoveries in theoretical biophysics come together in our work. As mentioned above, there is an impossibility theorem that states that, using observational data alone, the directed causal graph can be identified only up to its Markov equivalence class[35]. In this work, we aim to learn the undirected version of the causality graph, which another theorem[36] guarantees it belongs to the Markov equivalence class of the original graph, under the condition that there are no "unshielded confounders". In an independent line of development in biophysics, Milo et al.[37] showed that the structure of biological networks, such as gene regulation networks, is not random, but there are "(feedforward) motifs" that occur much more frequently than randomly. Despite different terminology across disciplines, both "unshielded confounders" in statistics, and "motifs" in biophysics refer to the same graph. In other words, empirical biophysical discoveries imply that the undirected graph we aim to learn is in the Markov equivalence class of the directed graph.

Celcomen uses the undirected underlying graph identified during training to bound and constrain its counterfactual predictions when generating synthetic data. In the generation module, called Simcomen (Simulated Communication Energy), we produce new samples in an adversarial approach by finding the most probable spatial transcriptomics sample under the joint distribution given the identified gene-gene forces and knocked out genes in cells, $p(\{s_{i \neq j}^\alpha\}|s_j^\beta = 0, g_{\alpha\beta}, g'_{\alpha\beta})$.

## Model architecture

To construct Celcomen, we used the Python packages of "PyG" (PyTorch Geometric),"PyTorch" and"scikit-learn". Celcomen uses two types of input: one encodes the graph structure, in the form of an adjacency matrix J for the k-NN graph created from the spatial locations of the cells and the hyperparameter n-neighbors; while the second is the log-transformed and then spherically normalized gene expression (such that the L2 norm of the gene expression vector of each cell is equal to 1). The hyperparameter n-neighbors should be chosen based on what information we want to disentangle in our system (see also Discussion above). By default, it is set to 6, which means that Celcomen tries to disentangle the intra-cellular gene regulation from the regulation of the cell's gene expression by its 6 nearest neighbors (for more details see Supplementary Fig. 5). Celcomen consists of one GCN layer (from the PyG implementation) with weights G′ and both input and output dimensions equal to the number of genes; and one linear layer from the PyTorch implementation with weights G and where again both the input and output dimensions are equal to the number of genes. Celcomen uses these two layers to construct the node embedding of node n as $Z = JSG' + SG$, and the loss function is calculated by $L = \log(z_{MFT}) - zmft\_scalar\ Tr(ZS^T)$, where $z_{MFT}$ is the mean-field approximation, and"$zmft\_scalar$" is a hyperparameter that adjusts the weighting of the mean-field approximation to ensure the loss function is lower bounded even when the mean-field approximation assumption is violated in the data. This hyperparameter should be chosen between 0 (big adjustment) and 1 (no adjustment, i.e., the mean field approximation is exact and not an approximation) such that the training process stably converges to a minimum (see also Supplementary Fig. 6).

## Spatial transcriptomics dataset curation and preprocessing

The fetal spleen datasets were curated from https://developmental.cellatlas.io/fetal-immune[20] in log-normalized form, which explicitly indicates log-transformation and library size normalization. The glioblastoma dataset was curated from 10x genomics at https://www.10xgenomics.com/datasets/ffpe-human-brain-cancer-data-with-human-immuno-oncology-profiling-panel-and-custom-add-on-1-standard and subjected to the same library size normalization, counts per million (CPM), and log-transformation, with a base of e; additionally, only genes that were expressed in at least 100 cells were kept. Due to the large size of the Xenium slide, a random square portion of the slide was chosen for analysis, this section is defined as cell centroid x-component > 6500 and <7000 and cell centroid y-component > 8000 and <8500. The entire fetal spleen slide was kept for each fetal slide sample as they are comparatively smaller than the original Xenium slide and post down-sampling, approximately the same size as the analyzed Xenium section. All data normalization were done using Scanpy (v1.9.8) in Python (v3.9.18)[38].

## Simulations testing Celcomen's predictions

Simulations were done in Python and completed by first generating a ground truth gene-gene interaction matrix. This was achieved by creating a n-genes by n-genes matrix of random values; for these experiments four genes were used. We then utilized Celcomen's generative module, Simcomen, to learn a spatially-resolved counts matrix reflective of the ground truth gene-gene interaction matrix. Comparisons to the randomly initialized count matrix are termed "Raw input" and those to the learned count matrix are termed "SCE output". To interrogate for self-consistency, we initialized Celcomen's inference module with a random gene-gene interaction matrix and asked it to utilize the learned count matrix from Simcomen to decipher the ground truth gene-gene interaction matrix. Comparisons to the Celcomen outputted gene-gene interaction matrix are termed "CCE output". Spearman correlation was used to compare the ground-truth gene-gene interaction values and the simulated-then-inferred gene-

gene interaction values to test for model robustness and identifiability. For all exact parameter values utilized during the experiments, see the "analysis.simulations.ipynb" notebook in the reproducibility GitHub.

## Biological testing of Celcomen's predictions

Biological testing of Celcomen's identifiability guarantee was done by training two Celcomen inference module instances at the same time and comparing their derived gene-gene interaction results. The first model instance, which we call sample-specific, was trained only on one sample. The second model instance, which we call rest, was trained on the remaining samples. Thus, these two model instances are never trained on the same samples. Each model is trained to completion utilizing the same model hyperparameters, and their gene-gene interaction matrices are retrieved after the final epoch. We correlate a flattened version of their gene-gene interaction matrices using Spearman's correlation as rank-based comparisons can most accurately capture an overall monotonic transformation of the values between these matrices. We repeat this experiment for each of the samples in the fetal spleen dataset. The results across each sample's experiments are aggregated together and compared in a bar plot. We derived a "random" control to compare to by shuffling the order of the flattened gene-gene interaction matrices and computing a correlation of the shuffled values. Mann–Whitney U test is used to derive p-values and all p-values are labeled on plot. For the full code utilized, see the "analysis.biological.ipynb" notebook in the reproducibility GitHub.

## Interferon knockout experiment on Xenium of human glioblastoma

Processed Xenium data was subjected to the inference module of Celcomen, CCE, and then these gene-gene interaction values were annotated as containing cytoplasmic, surface membrane (plasma membrane GO ID via GO cellular component), or secreted (extracellular space GO ID also via GO cellular component) genes according to their GO IDs from QuickGO[39]. *IFITM3* was knocked out in a randomly selected previously *IFITM3* positive cell. First neighbors were defined as less than 15 μm away and second neighbors were defined as less than 30 μm away. Changes in each gene's expression in each cell were calculated and these changes in expression pre- and post- perturbation were compared between different specified cellular subsets. These are the differential genes later used for differential expression analysis and pathway enrichment. Gene set enrichment analysis (GSEA) in R (v4.1.2) was utilized to perform pathway enrichment analysis on differentially post-perturbation affected genes. The interferon signature was derived directly from tissue by computing the differentially expressed genes between interferon high and low cells and taking the top 25, excluding the perturbed *IFITM3* as that would bias analyses. For the full model parameters and code utilized, see the "analysis.perturbation.ipynb" notebook in the reproducibility GitHub.

## Counterfactual prediction validation via in vivo perturbed lung tumors

Spatial perturbation data was acquired from previously published Perturb-map technology, GSE193460[27]. Their processed spaceranger output and annotations were read in and lesions not specific to a one gene KO (KP), as previously annotated, were identified and any spots that were within two degrees of a perturbation specific cluster were trimmed away; this was done via a < 100 filter in spatial distance with the value of 100 visually acquired from a histogram of spot-spot spatial distances (i.e., distance of 100 was the second non-zero peak). Lesions were then fed into the Celcomen model to identify gene-gene relationships and the trained gene-gene interaction matrix was used by Simcomen for counterfactual predictions. In detail, each lesion was examined for Tgfbr2$^+$ spots and had a random positive spot knocked out (KO) in terms of Tgfbr2 expression. Simcomen then utilized the learned gene-gene interaction matrix to predict the whole

transcriptome of every spot post perturbation. We then compared the change in expression in the KO spot compared to KP spots. Spearman correlation was used to compare model Tgfbr2 KO versus KP gene rankings with those directly derived from experimental Tgfbr2 KO spots and KP, i.e., the published data includes an in vivo bona fide Tgfbr2 KO lesion and this was used as ground truth. We derived "random" controls for each lesion by computing correlations on shuffled gene rankings of the observed and predicted differentials between Tgfbr2 KO and KP. Mann–Whitney U test is used to derive $p$-value when comparing observed lesion derived gene rankings with those from random shufflings. For the full code utilized, see the "analysis.biological.ipynb" notebook in the reproducibility GitHub.

## Reporting summary

Further information on research design is available in the Nature Portfolio Reporting Summary linked to this article.

## Data availability

Data analyzed in this manuscript is previously published and is available from https://developmental.cellatlas.io/fetal-immune for human fetal spleen Visium, from https://www.10xgenomics.com/datasets/human-lymph-node-1-standard-1-1-0 for human lymph node Visium, from https://www.10xgenomics.com/datasets/xenium-human-brain-preview-data-1-standard for healthy human brain Xenium, from https://www.10xgenomics.com/datasets/ffpe-human-brain-cancer-data-with-human-immuno-oncology-profiling-panel-and-custom-add-on-1-standard for human brain cancer Xenium, from https://www.ncbi.nlm.nih.gov/geo/query/acc.cgi?acc=GSE193460 for mouse perturbed lung tumor Visium[20,40]. Source data are provided with this paper.

## Code availability

Celcomen is available as a python package under the GPL-3.0 license at https://github.com/Teichlab/celcomen[41]. The code required for reproducing the analyses in this paper are at https://github.com/stathismegas/celcomen_reproducibility[42]. The specific version of the code associated with this publication is archived in Zenodo and is accessible at https://doi.org/10.5281/zenodo.17877584 and https://doi.org/10.5281/zenodo.17877629.

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

## Acknowledgements

We would like to thank Erick Armingol, Dinithi Sumanaweera, Maciej Wiatrak and Qingyuan Zhao for helpful feedback and discussions. This work was made possible in part by the Wellcome Trust (220540/Z/20/A). This research was funded in whole, or in part, by the Wellcome Trust [203151/Z/16/Z, 203151/A/16/Z] and the UKRI Medical Research Council [MC_PC_17230]. SM wants to thank Schmidt Sciences for their funding. For the purpose of open access, the author has applied a CC BY public copyright licence to any Author Accepted Manuscript version arising from this submission. The funders had no role in study design, data collection and analysis, decision to publish, or preparation of the manuscript. Figure cartoons were created with BioRender.com.

## Author contributions

S.M. conceived the theory of the model, implemented it in code and as a GitHub package, and wrote and edited the manuscript. D.G.C. performed the benchmarking on simulations and real data, the perturbation modeling and biological interpretation, and wrote and edited the manuscript. K.P. helped create Conda environments for the project. M.E., C.S., and K.P. gave feedback on the project. H.A. helped improve the GitHub repository. S.A.T. conceived the idea of disentangled representations in spatial data, provided key biological interpretation, and supervised the project.

## Competing interests

In the past three years, S.A.T. has received remuneration for scientific advisory board membership from Sanofi, GlaxoSmithKline, Foresite Labs and Qiagen. S.A.T. is a co-founder and holds equity in Transition Bio and Ensocell. From January 8th of 2024, S.A.T. is a part-time employee of GlaxoSmithKline. The remaining authors declare no competing interests.
