## [Transparent Peer Review file · Nature Communications]

Celcomen: spatial causal disentanglement for single-cell and tissue perturbation modeling

Corresponding Author: Professor Sarah Teichmann

Version 0:

Reviewer comments:

Reviewer #4

(Remarks to the Author)

The editor asked to comment on the author's response to original reviewers' concerns. Generally, I think the authors made revisions efforts based on the reviewers' concerns, while we have the following concerns to be addressed.

(1) Regarding the causal identifiability, which is the main point issue raised by the original NBT reviewers, I think it is still not explained or proved clearly. They equate $p(s|g, g')$ under two parameter values and conclude $g=h$ and $g'=h'$. That is statistical parameter identifiability. Causal identifiability is about recovering interventional quantities from observational data and needs assumptions such as the correct causal graph, no unobserved confounding, positivity, and often do calculus. None of that is addressed.

(2) The overall architecture of this paper is messy and makes it hard to read and understand. Actually, the manuscript they submitted to ICLR is much better. Some phrases need to be explained clearly. I suggest the authors to re-formulate the whole structure as a routine NC format including Background introduction, Results, Conclusion and Discussions. Some contents are directly copied from supplementary information. The author should reorganize the methods part and supplementary information very carefully.

(3) The validation of the counterfactual prediction is very limited, considering that there indeed only limited tissue perturbation data existed, while this limitation make the whole validation parts in-solid.

(4) A lot of notions are not explained clearly, making the whole paper very hard to be understand:

- a) In line 237-239, "Celcomen estimates for each spatial transcriptomics sample $\{s^i\}: \alpha = 1, \dots, G; i = 1, \dots, M\}$, where N is the number of cells and G is the number of cells in the sample". What's the difference between N and G ?
- b) What's the meaning of disentangled representation of gene interactions? Is that representing the links between genes or just embedded gene-gene interaction into a vector space?
- c) The author seems like using "force" to represent gene-gene interaction. What's exact meaning of gene-gene interaction forces?
- d) In line 301, author confirmed Celcomen can not infer more fine grained than the Markov equivalence class of graph in simulations. So what's the benefit of Celcomen compared to other methods.

In summary, although the authors made good attempts to revise the manuscript according to the reviewer's comments, the causal identifiability issue is still not clearly solved, and the whole manuscript is needed to be re-organized and extensively revised to fit NC.

(Remarks on code availability)

/

Version 1:

Reviewer comments:

Reviewer #4

(Remarks to the Author)

Most of my earlier concerns have been adequately addressed. I do have one new question: the authors' interpretation of "KP" lesions in the Perturb-map dataset may be problematic. In the original Perturb-map paper, "KP_x-y" clusters are described as "KP lesions not specific to a particular gene KO", rather than lesions without any KO. Please clarify it.

Minor concerns

1. Please correct a few typographical errors in the SI (e.g. "simpler simpler loss function" → "simpler loss function"; "adjectment" → "adjustment"), and standardize the naming of the generative module (e.g. consistently use "Simcomen (SCE)" rather than alternating between "Simcomen", "SCE", and "Simulated Communication Energy").

2. Some figure labels and legends are ambiguous. For example, in Fig. 1C the role of the SCE output and abbreviations such as "CCC output" and "SCC output" are not clearly defined and do not fully match the axis label and caption. I recommend harmonizing labels, legends and captions, and explicitly defining any abbreviations.

(Remarks on code availability)

REVIEWER COMMENTS

Reviewer #4 (Remarks to the Author):

The editor asked to comment on the author's response to original reviewers' concerns. Generally, I think the authors made revisions efforts based on the reviewers' concerns, while we have the following concerns to be addressed.

General response:

We thank Reviewer 4 for helpful comments and suggestions that allowed us to revise our manuscript and remove wording that could be confusing to the readers. We also thank Reviewer 4 for acknowledging our efforts in addressing Reviewer 3's previous comments. In particular we want to note that we performed all of the intricate simulations asked by Reviewer 3 and the results demonstrate that we 1) maintain state of the art performance against competitors in the causal structure learning task (i.e. recovering edges), while 2) for the first time offer the novel ability to in-silico simulate the whole-tissue response to genetic perturbations of one of more cells, 3) develop a way to validate our predictions in public in-vivo spatial genetic screens. In particular these last two contributions are the main novelties we introduce.

All new changes are highlighted **green** in the main text. We also have retained the **yellow** highlight of the changes we made to address the 3rd reviewer's comments.

(1) Regarding the causal identifiability, which is the main point issue raised by the original NBT reviewers, I think it is still not explained or proved clearly. They equate $p(s|g, g')$ under two parameter values and conclude $g=h$ and $g'=h'$. That is statistical parameter identifiability. Causal identifiability is about recovering interventional quantities from observational data and needs assumptions such as the correct causal graph, no unobserved confounding, positivity, and often do calculus. None of that is addressed.

Reply 1: We thank the reviewer for giving us the opportunity to clarify potentially confusing wording regarding identifiability guarantees in our work.

We want to start by raising a distinction between identifying the causal graph and identifying the counterfactuals. As mentioned in Squires and Uhler¹: " In some special, well-studied settings, background knowledge and human reasoning can be used to propose plausible directed graph models. However, in large systems such as gene regulatory networks, the directed graph is not known a priori, making it necessary to develop methods for learning the graph from data. Once this graph is learned, it can be

¹ Squires, C. & Uhler, C. Causal Structure Learning: A Combinatorial Perspective. *Found. Comput. Math.* **23**, 1781–1815 (2022).

used to predict the effects of interventions or distributional shifts, in contrast to traditional machine learning methods which can only make predictions on inputs that come from the same distribution as the training data.”

Indeed, our aim here is to *learn* from observational data the graph of how genes influence each other within cells and across cells, and then use the learnt graph to perform counterfactual predictions where we intervene on one or more cells. This aim is mathematically restricted by an impossibility theorem² saying that using only observational data the directed graph of these interactions can be only partially identified – essentially only its undirected version can be identified (for a more mathematically precise statement see next paragraph). Since, the causal graph cannot be identified exactly, then our counterfactuals can be identified only within a range, and not perfectly. This is why in our paper we never talk about counterfactual identifiability, but rather of identifiability of the undirected causal graph. This is why we do not need to assume what the underlying graph is, because we try to learn it. Moreover, this is also why we have not used do-calculus in our identifiability result, because we identify the graph. To clear any potential confusion to the readers we have now amended our text in 3 different places (including in the abstract) to replace the word “identifiability” with the more specific and clear phrase “**identifiability of causal structure**” (lines 22, 52, 77).

As the reviewer notes, we have indeed assumed “causal sufficiency”, also known as no unobserved confounding, and we quote here from our manuscript “**The entire neural network architecture of Celcomen is derived from its three main assumptions that: [...] 3) Causal sufficiency that there are no unobserved confounders that affect two or more genes.**”

At a more technical level, there is a nice story of how theorems in mathematics and discoveries in theoretical biophysics come together in our work. The impossibility theorem mentioned above states that, using observational data alone, the directed causal graph can be identified only up to a class of graphs, called the Markov equivalence class of the true graph. In this work, we aim to learn the undirected version of the causality graph, which another theorem³ guarantees it belongs to the Markov equivalence class of the original graph, under the condition that there are no “unshielded confounders” (see Response Figure 1a,b). In an independent line of development in biophysics, Milo et al⁴ showed that the structure of biological networks, such as gene regulation networks, is not random, but there are “motifs” that occur much more frequently than randomly (see Response Figure 1b,c). In fact, despite different terminology across disciplines, both “unshielded confounders” in statistics, and “motifs”

² Squires, C. & Uhler, C. Causal Structure Learning: A Combinatorial Perspective. *Found. Comput. Math.* **23**, 1781–1815 (2022).

³ Meek, C. Relating Graphical Frameworks: Undirected, Directed Acyclic and Chain Graph Models. *Rep. C.* (1995).

⁴ Milo, R. et al. Network Motifs: Simple Building Blocks of Complex Networks. *Science* (80-.). **298**, 824–827 (2002).

in biophysics refer to the same graph (shown in Response Figure 1b). In other words, empirical biophysical discoveries imply that the undirected graph we aim to learn is in the Markov equivalence class of the directed graph. We have added this paragraph into our Methods section (lines 332-345).

We also want to note that the inspiration for Celcomen comes from theoretical physics which studies causal interactions between fundamental particles often using what in statistics would be called “statistical” and not “causal” graphical models. Nonetheless these models move beyond correlations to uncover the direct causal effect of one particle to another, by developing tools that account and remove all possible confounders or other mechanisms mediating indirect effects between the particles. It is in this way that we use the terms causal and causality in the text. This also explains our use of the word forces or interactions, which within the framework of theories of fundamental particles are the causes behind how systems evolve. Within the Lagrangian formulation of mechanics, forces are in fact equal to the Lagrange multipliers, which are the quantities Celcomen calculates through a graph neural network. We chose the words causes, forces, and interactions, as a more intuitive way of referring to “Lagrange multipliers” which might be a term unfamiliar to computational biologists to whom we address this paper. To also clear any confusion in our paper regarding different uses of the term “causal” across disciplines we have now replaced the term “Structural Causal Model” with “Causal Models” (lines 240-254). Moreover we have added information in the methods section “One important property of Celcomen, supported by Proposition 2, is its ability to reduce confusion from confounders and mediators when estimating causal effects. Even when two genes show a high correlation without a direct causal link, Celcomen’s gene-gene force term (the Lagrange multiplier) tends toward zero, helping the model to distinguish genuine causal interactions from indirect associations. To illustrate the scale of the challenge, consider that a typical cell may express on the order of 10^4 genes and interact with 10 neighbors; as a result, any direct effect between two genes could be masked by 10^5 indirect effects mediated through one gene, or $\sim 10^{10}$ indirect effects mediated through two genes, etc. Celcomen aims to identify the subset of direct effects that best explain the observed data while accounting for such infinite layers of mediation and confounding.” (lines 324-340).

(2) The overall architecture of this paper is messy and makes it hard to read and understand. Actually, the manuscript they submitted to ICLR is much better. Some phrases need to be explained clearly. I suggest the authors to re-formulate the whole structure as a routine NC format including Background introduction, Results, Conclusion and Discussions. Some contents are directly copied from supplementary information.

The author should reorganize the methods part and supplementary information very carefully.

Reply Q2: We thank the reviewer for suggestions that we fully agree would strengthen the paper and its presentation. While transferring our submission from Nature Biotechnology to Nature Communications we chose to keep the format the same to make it easier for reviewer 3 to keep track of the changes. Again, to have fewer moving parts while the paper is under review, we have maintained the format in this submission round as well. However, we agree with the reviewer, and remain committed to work with the editing team to change the text into an optimal format prior to publication.

(3) The validation of the counterfactual prediction is very limited, considering that there indeed only limited tissue perturbation data existed, while this limitation make the whole validation parts in-solid.

Reply Q3: We thank the reviewer for raising the issue of further validating the model while acknowledging that there is a very limited number of in-vivo spatial genetic screens that have profiled the full transcriptome. Full-transcriptome profiling is important to abide by our assumption of no unobserved confounding, as the reviewer correctly pointed out in his first comment. We therefore choose to leave further validation of Celcomen and its extensions as future work.

(4) A lot of notions are not explained clearly, making the whole paper very hard to be understand:

a) In line 237-239, "Celcomen estimates for each spatial transcriptomics sample $\{s\}$: $\alpha = 1, \dots, G; i = 1, \dots, N\}$, where N is the number of cells and G is the number of cells in the sample". What's the difference between N and G ?

b) What's the meaning of disentangled representation of gene interactions? Is that representing the links between genes or just embedded gene-gene interaction into a vector space?

c) The author seems like using "force" to represent gene-gene interaction. What's exact meaning of gene-gene interaction forces?

d) In line 301, author confirmed Celcomen can not infer more fine grained than the Markov equivalence class of graph in simulations. So what's the benefit of Celcomen compared to other methods.

Reply Q4:

We thank the reviewer for identifying important typos as well as giving us the opportunity to comment on these and improve the text accordingly.

a) We thank the reviewer for pointing out this typo that would have caused confusion to readers. We have corrected it: "G is the number of **genes** in the sample".

b) The disentanglement we are talking about here is the disentanglement of the causes into intracellular and intercellular ones. Once these have been identified, Celcomen calculates a representation of each cell $Z=Z_{intra}+Z_{inter}$, which means that the disentanglement of causes also leads to a disentangled representation of each cell.

c) As we also mention above (and reproduce here some of the response), the inspiration for Celcomen comes from theoretical physics which studies causal interactions between fundamental particles. This explains our use of the word forces or interactions, which, within the framework of theories of fundamental particles, are the causes behind how systems evolve. Within the Lagrangian formulation of mechanics, forces are in fact equal to the Lagrange multipliers, which are the quantities Celcomen calculates through a graph neural network. We chose the words causes, forces, and interactions, as a more intuitive way of referring to “Lagrange multipliers” which might be a term unfamiliar to computational biologists to whom we address this paper.

d) We believe that the main contribution of Celcomen is not in the field of statistics, but rather in spatial genomics. In particular we introduce the notion of spatial counterfactuals to the field of spatial biology and leverage it to create (to our knowledge) the first perturbation model of virtual tissues and introduce an approach of validating it on spatial perturbation screens – akin to how perturbation models of virtual cells are validated on dissociated perturbation screens (such as perturb-seq). However, another important aspect of this model is the mathematical theory behind it and the identifiability guarantees regarding the causal structure, which is explicitly leveraged to predict our spatial counterfactuals.

We thank the reviewer for their kind recognition of our revisions. We are optimistic that we have now addressed the reviewer's remaining concerns and are excited to release this work to the public. Please find below the aforementioned point-by-point response:

Comment: *Most of my earlier concerns have been adequately addressed. I do have one new question: the authors' interpretation of "KP" lesions in the Perturb-map dataset may be problematic. In the original Perturb-map paper, "KP_x-y" clusters are described as "KP lesions not specific to a particular gene KO", rather than lesions without any KO. Please clarify it.*

Response: We appreciate the reviewer for highlighting this concern. We apologize for the mistake, the language of "wild-type" is incorrect and indeed should be correctly described as lesions not specific to a given gene KO. This has now been corrected throughout the manuscript, see highlighted areas. In brief, we have renamed wild-type (WT) lesions to lesions that are not specific to a given gene KO (abbreviated as KP). This approach is similar to what was performed in the original Perturb-Map paper which compared their lesions specific to one gene KO (e.g. *Tgfbr2* KO) to those that were not specific to a given gene KO (what we have now labeled as KP not WT); for example, see Fig. 5D in their manuscript. This textual change does not change the results presented in this paper. See an example change as follows, "The ideology behind this in vivo validation was to 1) train on lesions not specific to one gene (KP), 2) simulate a one gene specific genetic perturbations in KP tissue, 3) compare model predicted transcriptomic differences with experimentally observed differences." (page 4 lines 171-174).

Comment: *1. Please correct a few typographical errors in the SI (e.g. "simpler simpler loss function" → "simpler loss function"; "adjectment" → "adjustment"), and standardize the naming of the generative module (e.g. consistently use "Simcomen (SCE)" rather than alternating between "Simcomen", "SCE", and "Simulated Communication Energy").*

Response: We thank the reviewer for highlighting these typographical errors. We have now corrected them all and appreciate the reviewer for their help on this matter.

Comment: *2. Some figure labels and legends are ambiguous. For example, in Fig. 1C the role of the SCE output and abbreviations such as "CCC output" and "SCC output" are not clearly defined and do not fully match the axis label and caption. I recommend harmonizing labels, legends and captions, and explicitly defining any abbreviations.*

Response: We apologize for this typographical error. We have now corrected this and thank the reviewer for helping to improve our manuscript.